# SDGym: Low-Code Reinforcement Learning Environments using System Dynamics Models

## Abstract

Understanding the long-term impact of algorithmic interventions on society is vital to achieving responsible AI. Traditional evaluation strategies often fall short due to the complex, adaptive and dynamic nature of society. While reinforcement learning (RL) can be a powerful approach for optimizing decisions in dynamic settings, the difficulty of realistic environment design remains a barrier to building robust agents that perform well in practical settings. To address this issue we tap into the field of system dynamics (SD) as a complementary method that incorporates collaborative simulation model specification practices. We introduce SDGym, a low-code library built on the OpenAI Gym framework which enables the generation of custom RL environments based on SD simulation models. Through a feasibility study we validate that well specified, rich RL environments can be generated from preexisting SD models and a few lines of configuration code. We demonstrate the capabilities of the SDGym environment using an SD model of the electric vehicle adoption problem. We compare two SD simulators, PySD and BPTK-Py for parity, and train a D4PG agent using the Acme framework to showcase learning and environment interaction. Our preliminary findings underscore the dual potential of SD to improve RL environment design and for RL to improve dynamic policy discovery within SD models. By open-sourcing SDGym, the intent is to galvanize further research and promote adoption across the SD and RL communities, thereby catalyzing collaboration in this emerging interdisciplinary space.

## 1 Introduction

At the heart of the pursuit of responsible AI is the ability to proactively understand and evaluate the potential impact of an algorithmic intervention on society. This is especially important for populations that are disproportionately impacted when ML systems amplify or propagate harmful societal biases. Due to the dynamic, complex and adaptive nature of society, evaluations based on static settings are not sufficient to provide insight on the long-term impacts of algorithmic interventions (Liu et al., 2018). Reinforcement learning (RL) has emerged as a promising technique to evaluate the long-term behavior and impact of algorithms (D'Amour et al., 2020). Creating rich RL environments that accurately reflect real-world experiences, domain expertise and socio-cultural factors thus becomes central to this goal. RL environment design is crucial to an agent's training, its ability to generalize and the risk of potential overfitting (Cobbe et al., 2019; Zhang et al., 2018a; Reda et al., 2020). However, the challenging nature of environment design can sometimes lead to overly simplified environment models. Relying on such models can lead to a false sense of robustness and fairness, resulting in potentially bad outcomes when agents are deployed into society.

Recognizing this, there's a growing interest in leveraging simulation models to improve the richness and relevance of RL environments (Leon et al., 2023; Sivakumar et al., 2022; Belsare et al., 2022). System Dynamics (SD) offers an approach to create dynamic hypotheses and simulation models of complex problems (Forrester, 1958), making it an attractive tool for RL environment design. Community-based Systems Dynamics (CBSD) further extends SD by taking a participatory approach to building comprehensive models through inclusive group model-building, system analysis and problem solving (Hovmand, 2013; Martin Jr et al., 2020).

The widespread availability of SD models across diverse domains presents an untapped opportunity (Zanker et al., 2021). If these SD models can be seamlessly integrated into RL environments, it would pave the way for more comprehensive evaluations of algorithms and facilitate interdisciplinary collaboration. Moreover, integrating RL with SD models can accelerate the process of identifying optimal policy interventions for complex societal challenges (Ghaffarzadegan et al., 2011). Integrating these two modeling methodologies is not without challenges however. Achieving this mutually beneficial representation goal requires overcoming a number of perceived and real incompatibilities between the two methods of problem modeling.

Our primary contribution is showcasing the integration of SD models into the RL setting through a feasibility study and proof-of-concept in which we initialize an RL environment using a sample SD model. This work takes a multidisciplinary approach to addressing gaps in RL environment design. Next we highlight two areas for further research: 1) ML fairness and robustness using RL environments based on SD models, and 2) the application of RL in areas like intervention discovery, policy and calibration optimization of SD models. Finally, to inform further research efforts and promote adoption across the SD and ML communities we have open-sourced *SDGym*, a flexible library built on the OpenAI Gym (Brockman et al., 2016) to enable reproducibility of results in this paper and support future extensibility[1].

## 2 Background

### 2.1 System Dynamics

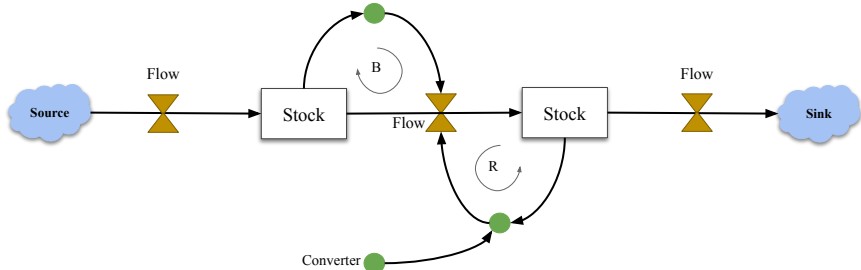

Figure 1: Illustrative diagram showing different components and notations of SD framework

SD is as a problem-oriented framework for helping humans understand and make decisions in complex dynamical systems characterized by interdependent variables and feedback loops (Hovmand, 2013). A causal loop diagram (CLD) provides a qualitative visual representation of causal relationships in a system, while a stock and flow (SFD) simulation model - shown in Figure 1 - is quantitative mathematical representation of a system. Stocks are accumulations in the system, and flows are rates of change in the levels of stocks with respect to time. Auxiliary variables such as converters act as either constants or intermediate functions that influence the system's dynamics. Feedback loops are cyclical chains of causal relationships that can either be reinforcing (positive feedback, R) or balancing (negative feedback, B), working towards maintaining a state of equilibrium. In SD, a simulation model is constructed to emulate the real-world system in question. This model is used by stakeholders for policy analysis, sensitivity tests and hypothesis design and testing.

### 2.2 Reinforcement Learning

RL is a machine learning approach focused on automated decision-making using interaction with an environment. The primary objective in RL is for agents to learn optimal strategies or policies that maximize a cumulative reward over time. This learning process is driven by the feedback loop between the agent's actions and the environment's responses. In a typical RL setup, an agent interacts with a dynamic environment over discrete time steps $\mathcal{T}$. At each time step $t$, the agent observes the state $s_t$ of the environment, which can be either fully or partially observed. Based on this observation and its current policy $\pi$, which maps states to distributions over possible actions $\mathcal{A}$, the agent selects an action $a_t$. This action transitions

---

[1]The code will be made available on acceptance.

the environment to a new state $s_{t+1}$ and results in a reward $r_t$. The reward function $R$ provides feedback to the agent about the quality of its actions. The overarching goal of the agent is to learn an optimal policy $\pi^*$ that maximizes the expected cumulative reward over time.

There are several ways to categorize the methods and approaches in RL, highlighting the different strategies researchers use to address decision-making problems.

### 2.2.1 Online vs Offline RL

The RL paradigm can be broadly categorized into online and offline approaches based on agentś interaction with the environment. In online RL, the agent learns by actively interacting with the environment in real-time. It continuously updates its policy based on the immediate feedback it receives from the environment. In contrast, offline RL, also known as batch RL, involves the agent learning from a fixed dataset of past interaction without further exploration. This approach is particularly useful in scenarios where online exploration is either expensive or risky (Levine et al., 2020).

### 2.2.2 Model-based vs Model-free RL

Another important distinction in RL is between model-based and model-free approaches, which pertains to the agentś knowledge and use of the environment's dynamics. With model-based methods the agent learns or has access to a model of the environment's dynamics. This model predicts the next state and reward given the current state and action. The agent uses this model to plan and make decisions. The advantage of this approach is that it can be more sample-efficient, as the agent can learn from simulated experiences (Moerland et al., 2023). On the other hand, model-free methods do not rely on an explicit model of the environment. Instead, they directly learn the value function or policy from interactions with the environment. While potentially requiring more samples, model-free methods can be more robust in environments with complex or unknown dynamics.

### 2.2.3 On-policy vs Off-policy

In addition to above, there are two predominant learning paradigms in RL: on-policy and off-policy. On-policy methods entail that the agent learns directly from its current exploration strategy. Conversely, off-policy methods allow the agent to learn from a previous strategy while exploring with a different one, introducing an element of flexibility in the learning process.

## 2.3 Opportunities for Integration

The complementary nature of RL and SD opens up possibilities for synergistic approaches to problem solving and decision making. SD modeling offers a robust framework for environment design, particularly in RL scenarios. By employing participatory approaches like CBSD, SD facilitates stakeholder engagement that accelerates the development of more comprehensive simulation models. This inclusive strategy tends to be effective at eliciting nuanced insights which can lead to a more accurate representation of complex system interactions. When used in RL, the improved environment representation can lead to better state transition dynamics, making the RL agent's exploration and exploitation processes more informed and robust. Furthermore, as SD democratizes the modeling process by lowering barriers, it facilitates quicker iterations and refinements of these models. This tighter feedback loop can result in better maintained RL environments that can stay up-to-date with real-world changes. Lastly, the structured simulation models with exogenous variables and accumulators found in SD offer a robust mechanism for generating multiple scenario-based environment conditions. Such flexibility becomes indispensable when dealing with domains like climate change and sustainable development goals, which necessitate the exploration of a multitude of scenarios.

## 2.4 Paradigm Differences

RL and SD have grown in separate disciplines with seemingly incompatible paradigms, however we observe they offer complementary frameworks. In Table 1 we highlight the most important differences in paradigms

that we aim to reconcile with our work. Note that these represent the traditional approaches, and with some effort these differences can or have been addressed albeit independently.

Table 1: A comparison of the differences between the SD and RL paradigms

| Aspect | Reinforcement Learning (RL) | System Dynamics (SD) |
|---|---|---|
| Learning process | Autonomous agents explore and exploit to optimize an objective. | Humans learn to make controlled interventions based on domain expertise. |
| Control frequency | Actions and feedback at discrete time steps. | Changing initial conditions and observing over the entire time horizon. |
| Stability | Environmental dynamics are predefined to focus on optimization problems. | Interventions include changes to the model structure for "what-if" scenarios to test hypotheses. |
| Objective function | Clear objective function, often cumulative reward optimization. | Precise objective not required, focuses on system stability & behavioural patterns. |
| Data needs | Requires large data sets for effective exploration | Works with limited data due to domain knowledge and qualitative inputs. |

RL can benefit from the systemic insights provided by SD, especially in constructing more realistic and nuanced environments. Conversely, SD can leverage RL's computational methods to automate the search for optimal intervention strategies, thereby enhancing its utility in policy planning and decision-making. The inclusion of CBSD's participatory methodologies, anchored in community wisdom and lived experiences, can augment RL and ML by paving the way for the development of more equitable and societally grounded solutions.

While the integration of these diverse paradigms is likely to encounter theoretical and practical challenges, such frictions could in fact stimulate innovation in both fields.

## 3 Environment Design

### 3.1 Recasting SD for RL

Conventionally, SD allows a human agent to tweak initial conditions and observe the unfolding system trajectories over the entire simulation. We note that this 'single-shot' interaction is less of feature and more of a limitation, stemming from having a large space of possible interventions. To address these challenges, we frame the task of finding interventions or combination of interventions in the SD model as an RL problem, which would enable the agent to make dynamic interventions over the simulation period, rather than just at the start. Our problem framing for this approach is presented in Figure 2.

In this integrated framework, the state space $\mathcal{S}$ and dynamics are governed by the underlying SD model, while the RL agent engages with this environment. An agent observes the environment, intervenes with an action $a_t$ over discrete $t \in T$ time steps, and receives a reward $r_t$ for each action. The goal is to learn an optimal policy $\pi^*$ that reflects the best decision making strategy in a given state $s_t$ to maximize the cumulative reward.

We introduce SDGym, a low-code library to realize the integration approach described above. This library provides an interface to initialize an OpenAI Gym environment based entirely on an SD model. In the following sections we discuss the components of the library and design choices made.

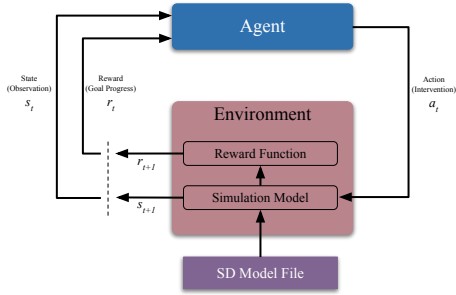

Figure 2: Mapping SD intervention discovery as an RL task

## 3.2 Environment

At the core of SDGym is the `SDEnv` class, which is derived from the `Env` class provided by the OpenAI Gym (Brockman et al., 2016). This environment is composed of a *simulation model*, a *state* object and a *reward function*. We provide a simple interface that can initialize an environment using an SD model file and optional parameters.

The `Params` class provides a simple configuration object for the environment. Configurable parameters include the model file path, initial conditions for stocks, and simulation start and stop times. We also support different environment time step and simulation time step configurations, for more granular simulation data. Additional parameters will be presented as we discuss environment components below.

## 3.3 Simulation Model

A fully defined simulation SD model is used to initialize the environment. We design an abstract `SDSimulator` class that interfaces with the simulation model. The simulator enables access and overrides to the models variables, in addition to injecting inputs during simulations. We provide implementations using PySD (Martin-Martinez et al., 2022; Houghton & Siegel, 2015) and BPTK-Py (transentis labs, 2018), two popular libraries for leveraging SD models in code. The `simulator` parameter is used to specify which implementation to use when initializing the environment.

We implement thin wrappers around the PySD and BPTK-Py, given they provide near complete interfaces for interacting with the SD model and simulation. Consequently, we inherit their limitations, impacting the coverage of SD models we are able to support. PySD supports the XMILE (for System Dynamics TC, 2015) and MDL (Systems) file formats, but with varying supported functionality[2]. BPTK-Py on the other hand only supports the XMILE file format, with some limitations[3]. A major limitation in BPTK-Py is its lack of support for dynamic input injection. We implement our own input injection functionality in the simulator to bridge this gap.

While both libraries are extensible to support missing features, workarounds could involve replacing the unsupported functions with corresponding static agent actions in most cases. In some cases however, models may have to be redesigned, or may not be supported without extensions to the libraries.

## 3.4 State Representation

The `State` class provides access to all variables in the SD model, including stocks, flows and converters. It also contains the simulation object in its current state, and a history of all previous and current observations. Observations of the agent can be limited using the `observables` parameter during environment setup. In accordance with RL principles, all actionable variables are excluded from observations. The observation space is simplified into a continuous space with infinite limits ($S \subseteq \mathbb{R}^m$), where $m$ is the number of observable variables. Additionally we support a `Dict` space representation for human readability.

---

[2]https://pysd.readthedocs.io/en/master/#limitations
[3]https://bptk.transentis.com/usage/limitations.html

### 3.5 Action Space

Actionable variables are SD model variables that can be injected with input throughout the simulation. We constrain inputs to converter variables that have constant values, as defined in the SD model. The `actionables` parameter can be used to further constrain the set of variables that accept input. We do not currently allow inputs to function or flow variables since they represent the dynamics in the system. Structural changes would cause instability, impacting the ability of an agent to learn effectively. Humans on the other hand are able to alter the the structure of the SD model for hypothesis testing, which an agent doesn't engage in. Future work could look into approaches for supporting generative actions that alter the structure of the environment, while improving the effectiveness of the learning process.

For type safety, an explicit mapping of SD units and Numpy types can be provided using the `sd_units_types` parameter, otherwise types are inferred using heuristics of the variable name. Min-max limits of actionable variables are read from the SD model. As a fallback the `default_sd_units_limits` parameter configures default limits per SD unit when variable limits are not set. Additionally the `sd_var_limits_override` parameters can be used to override limits per variable.

Interventions in SD models are designed to influence zero or more actionable variables, which is similar to the joint control problem in robotics (Liu et al., 2021). Thus we define an action space $\mathcal{A} = P(\bigcup V_n)$ which represents the power set of all input variables $V$. To generalize across problem settings we allow continuous, discrete, or categorical variables as inputs. We implement the action space (Zhu et al., 2022) as a union of all three variable types as shown in Equation 1, with examples in Table 2. For each action vector $a \in \mathcal{A}$ of length $n$, each input or action value $a_i$ has a specified range of valid values. For continuous and discrete variables, we define this range using minimum and maximum limits $(min_i, max_i)$. For categorical variables, the range consists of the indices of the available categories $(cat_i)$ for the input variable.

$$\mathcal{A} = \bigcup \begin{cases} \{a | a \in \mathbb{R}, \, min_i \leq a_i \leq max_i\}, \\ \{a | a \in \mathbb{Z}, \, min_i \leq a_i \leq max_i\}, \\ \{a | a \in \{0, \dots, |cat_i| - 1\} \end{cases} \tag{1}$$

Table 2: Action spaces in the composite action space

| Variable Type | Example | Gym Space | Numpy Type |
|---|---|---|---|
| Continuous | $[0, 1]$ | Box | float64 |
| Discrete | $\{1, \dots, 5\}$ | Box | int64 |
| Categorical | $\{2, 4, 5, 8\}$ | Discrete | int32 |

To facilitate agent learning, we implement a wrapper for the environment that flattens the composite action space into a single `Box` space of continuous variables and rescales to $[-1, 1]$, formalized in Equation 2. The process is reversed in the environment when actions are taken.

$$\mathcal{A} = \bigcup \{a \, | \, a \in \mathbb{R}, -1 \leq a \leq +1\} \tag{2}$$

While flattening the composite action space into a single continuous space facilitates the use of continuous-action RL algorithms, it introduces a potential limitation in terms of flexibility and precision. However, by reversing this process in the environment when actions are taken, the system can bridge the gap between the agent's continuous output and the environment's composite action requirements, mitigating some of the challenges but still necessitating careful consideration in the mapping strategy, particularly when employing discrete-action RL algorithms.

To enable more organic learning algorithms that aim to mimic SD-like intervention discovery, the environment supports a hybrid or parameterized action space (Neunert et al., 2020) through the

`parameterize_action_space` parameter. Enabling this introduces a meta-action that indicates whether to intervene on a particular variable. The input becomes a tuple $(x, a)$ where $x$ is a binary meta-action and the parameter $a$ is the intended input to the variable, as shown in Equation 3.

$$\mathcal{A} = \bigcup \{(x, a) \mid x \in \{0, 1\}, a \in \mathbb{R}, -1 \le a \le +1\} \tag{3}$$

**Terminology clarification:** In RL terminology, both continuous and discrete values (as understood mathematically) are often termed 'continuous', while 'categorical' values are labeled as 'discrete'. This contrasts with standard mathematical definitions. To ensure definition parity between SD and RL, we adhere to the mathematical terminology: 'continuous' for values within a range, 'discrete' for countable values with order, and 'categorical' for distinct categories without order. For clarity, the action space is available as a `Dict` space for human-readability.

### 3.6 Reward Function

We start with simple reward functions $r_t = R(s_t, a_t, s_{t+1})$ inspired by the ML Fairness Gym (D'Amour et al., 2020). The reward at time $t$, denoted as $r_t$, is based on the change in the state variable between the consecutive time steps $t$ and $t + 1$. A reward can be based on an increase or decrease in the value of a state variable, where the reward amount is the change in value. A variant of the reward function binarizes the reward amount in order to regularize the signal to the agent. This binary function yields 1 if the state variable shows an increase or decrease and 0 otherwise, as outlined in Equation 4. During environment reset, baseline values are established from the state variables.

$$r_t = s_{t+1} - s_t \qquad\qquad binarized(r)_t = \begin{cases} 1 & \text{if } s_{t+1} - s_t > 0 \\ 0 & \text{else} \end{cases} \tag{4}$$

We support custom functions for computing rewards. While this is primarily for use in more complex reward scenarios, it is also useful for simple cases when the reward doesn't exist as a variable in the SD model and the model cannot be modified.

### 3.7 Control Agents

The architectural decisions behind SDGym are aimed at maximizing compatibility with a wide range of RL agents. Besides the constraint of using continuous action and observation spaces, no other specific limitations are enforced on the environment. This design choice promotes a broad applicability to most RL algorithms.

### 3.8 Implications for SD model design

It's important to note that SD models were not designed for interaction with RL agents. As a consequence, certain model adjustments may be necessary to ensure RL compatibility. For example, as we discussed earlier, constraints inherent to SD libraries could impact what functions can be used in models. Additionally, due to a lack of support for functions and flow modifications in the action space, models would need to be redesigned to expose simpler variables that would work for input injection.

## 4 Using SDGym

To assess the utility of SDGym we set up an RL environment using the "Electric Vehicle Popularity in Norway" SD model from Orliuk & Yermolova (2019). In this work, an SD model was built to understand the Norwegian car market and the impact of government policy with respect to electric car usage over a ten year period. The model is of moderate size and complexity as seen in Figure 3. It captures a diversity of factors such as tax policy, car pricing, oil industry stakeholder worries, cost of electricity and fuel, oil reserves and

production, car lifetimes, charging stations, effect delays, stochasticity and more. This SD model is able to simulate the impact of policy interventions like a VAT tax exemption that drove adoption of electric cars in the past.

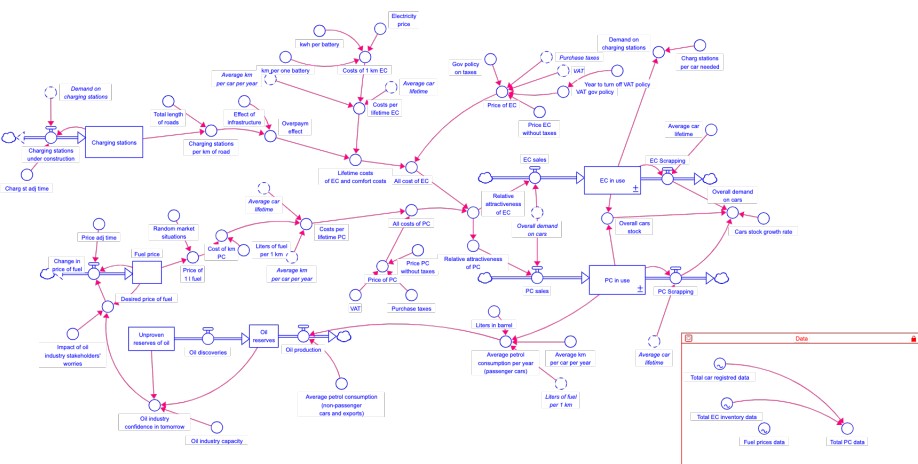

Figure 3: The "Electric Vehicle Popularity in Norway" SD model (Orliuk & Yermolova, 2019)

In the following sections we demonstrate basic capabilities of the SDGym library. We compare the simulators that can be used within the framework, and demonstrate the feasibility of agent-environment interactions and learning.

## 4.1 Comparing PySD and BPTK-Py Simulators

In the first demo we validate functional parity and simulation performance between the PySD and BPTK-Py simulators. The task is to determine if the two simulators yield consistent and equivalent results, while highlighting the limitations of each library.

### 4.1.1 Environment Setup

We configure a simple environment with the SD model, defaulting to all constant converters being actionable, and all other variables being observable. We do not define a reward function. For debuggability we do not flatten the action space or observation space. We parameterize the action space and set the meta-action to no-action for all action values. This allows us to test simulations of the models as is, without interventions. We initialize two variants of the base environment - one with the PySD simulator, and the other with the BPTK-Py simulator.

### 4.1.2 Discussion

We observe the same trajectories of state variables in both simulators in Figure 4. Simulation performance parity indicates that both simulators are equally capable of powering RL environments. Considerations for choosing one simulator over another would thus depend on its limitations (Section 3.3).

For example, PySD does not support the `RANDOM` function in XMILE files. This function is used in our SD model, so we implement two possible solutions. The more sustainable approach implements support in the PySD library, while a workaround replaces the variable with a random constant. We use the latter in our demos. We encounter a different limitation in BPTK-Py: non-negative stocks are not supported. This doesn't impact our model however, as a negative stock is an invalid state transition in this model, occurring only when we intervene with invalid inputs. We nonetheless explore a solution to implement support for non-negative stocks in the BPTK-Py library. Empirically we find PySD to be easier to extend compared to BPTK-Py.

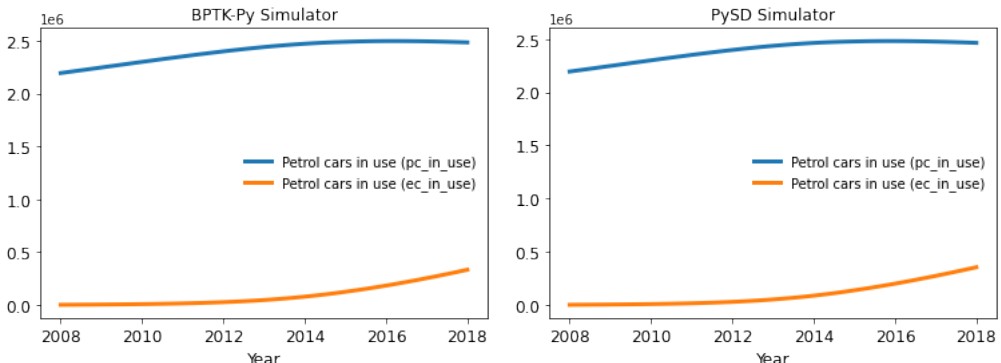

Figure 4: Trajectories of petrol cars and electric cars in use, using the BPTK-Py simulator (left) and the PySD simulator (right)

## 4.2 RL for SD Interventions

In the second demo we train an RL agent that can intervene on the environment to achieve a goal. We validate two key propositions: 1) well-specified RL environments can be effectively initialized solely based on an SD model, thereby expediting the environment construction phase, and 2) dynamic interventions can be learned for SD models using RL agents, resulting in more robust policies while saving time in intervention discovery.

### 4.2.1 Environment Setup

**Simulator.** Given parity in performance of simulators, we arbitrarily use the PySD simulator.

**Reward function.** We establish an initial objective aimed at amplifying the adoption of electric vehicles, aligning with historical governmental policies discussed by Orliuk & Yermolova (2019). We further refine this objective to focus on increasing the share of electric vehicles in the total car population. This specification is crucial, as the practical aim is to not only boost the utilization of electric vehicles but also decrease petrol cars.

We construct a scalar reward function using the `ScalarDeltaReward` object, setting the reward value based on the increase in the total share of electric cars. We implement a custom function with the state object in Equation 5 since there is no single variable in the model that reflects this value.

$$f(s)_t = \frac{ec\_in\_use_t}{ec\_in\_use_t + pc\_in\_use_t} * 100 \tag{5}$$

Here, $f(s)_t$ signifies the computed state variable to be used in the reward function at $t$, while $ec\_in\_use_t$ and $pc\_in\_use_t$ denote the number of electric and petrol cars in use at that time, respectively.

**Observation space.** The environment is fully observable. All non-actionable state variables are available to the agent.

**Action space.** We visually analyze the model using an abridged version of the Loops that Matter framework (Schoenberg et al., 2020) to identify candidate variables for intervention. We compose a set of actionable variables that indirectly impact petrol car usage and electric car usage as shown in Table 3. We reasonably assign variable types and set limits on variables so we can validate core features of SDGym. Since we are using an deep RL agent, we flatten and rescale the action space, as shown earlier in Equation 2.

Table 3: Actionable variables that constitute the action space

| Name | Variable Type | Valid Inputs |
|---|---|---|
| average_car_lifetime | Categorical | $\{1, 3, 5, 7, 10, 15\}$ |
| km_per_one_battery | Continuous | $[10, 1000]$ |
| kwh_per_battery | Continuous | $[10, 100]$ |
| electricity_price | Continuous | $[1, 10]$ |
| price_ec_without_taxes | Continuous | $[20000, 100000]$ |
| price_pc_without_taxes | Continuous | $[10000, 70000]$ |
| vat | Categorical | $\{0.15, 0.3, 0.44, 0.5\}$ |
| gov_policy_on_taxes | Continuous | $[0, 1]$ |
| oil_industry_capacity | Discrete | $\{1000000, \dots, 2000000\}$ |

**Agent.** We leverage the ACME library (Hoffman et al., 2020) to train a D4PG agent (Barth-Maron et al., 2018) that can learn a policy for achieving our specified goal. We choose D4PG due to its versatility in a variety of settings. As a control we implement a naive untrained agent using the same policy network setup as the trained agent.

**Training and evaluation parameters.** The simulation length is 10 years. We specify a time step of 0.1 (every 1.2 months), resulting in an episode length of 1000 time steps. We train the agent for 100 episodes, with a Gaussian noise sigma of 0.7 and a discount of 0.99.

### 4.2.2 Discussion

We observe that the D4PG agent performs better, albeit inconsistently given the length of training, compared to a naive untrained agent during evaluation, shown in Figure 5. With more training and tuning, we expect it to perform consistently better. Empirical evidence shows a dynamic strategy that adjusts interventions at various time steps. This is an improvement over intervention discovery performed by humans in SD models, which primarily focuses on changing initial conditions alone. Our results validate that a pre-defined SD model is viable for generating a complete RL environment that an agent can learn in. Additionally, RL is able to learn a dynamic intervention strategy in SD models given a goal. As with all RL settings, designing a good reward function is incredibly important to learning a good policy.

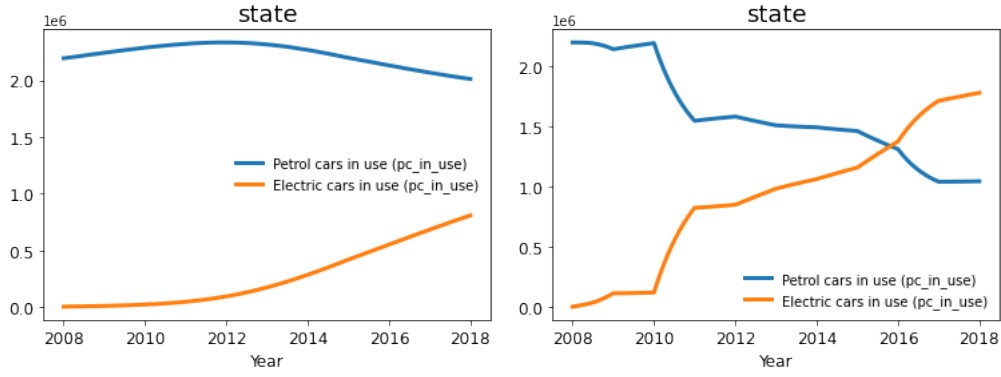

Figure 5: Trajectories of petrol cars and electric cars in use, driven by actions of a naive agent (left) and a D4PG agent (right)

### 4.3 Extensibility

We choose the Gym framework as it enables extensions of SDGym to leverage advances in the Gym ecosystem. For example, SDGym can be adapted for fairness evaluations using ML Fairness Gym (D'Amour et al., 2020) and multi agent RL can can explored using PettingZoo (Terry et al., 2021). As we demonstrated with ACME, any learning libraries compatible with the Gym framework can be used in any SDGym environments.

## 5 Related Work

Robustness is a significant challenge for deploying ML systems effectively (D'Amour et al., 2022; Dulac-Arnold et al., 2021). Various strategies have been proposed in understanding and addressing this problem. Pinto et al. (2017) enhance robust reinforcement learning by introducing destabilizing forces through an adversarial agent, while Wang et al. (2019) leverage evolutionary algorithms for generative environments while optimizing agent learning in the diverse conditions created. Both approaches primarily target aleatoric uncertainty in models through random perturbations. In contrast, methods to confront epistemic uncertainity have shifted towards causal strategies (Zhang & Bareinboim, 2020; Lee & Bareinboim, 2020). Our work resonates with these causal techniques, but with a distinct emphasis: we use SD simulation models as the foundational causal models. Instead of inferring causality from observational data, SD models hypothesize the data-generating process.

In the field of RL modeling, the challenge centers on environment design (Nisan & Ronen, 2001). The creation of new RL environments remains a relatively unexplored area, since most environments are manually designed and often reused. Evolutionary or generative approaches require existing environments or face generalization challenges (Wang et al., 2019; Zhang et al., 2018b). In RL, the shift towards model-based techniques, as highlighted by Ha & Schmidhuber (2018), addressed the challenge of extensive dataset requirements. Their proposed method centers on agents learning and utilizing an internal environment model for informed decision-making, enhancing efficiency by enabling predictive insights into future outcomes. This methodology finds parallel with the structured approach of SD models and learning through agent-environment interactions, SD models present a systematic framework for crafting RL environments, anchored in established system dynamics principles. The closest work to ours focuses on translation of other artifacts into an initial RL environment. Krimstein (2021) develops a methodology to generate simulation environments using machine-tool descriptions and Azad et al. (2021) introduces a specification for defining environments.

Broadening the lens, we explore a multi-disciplinary approach to understand cross-pollination of learning approaches and simulation modeling paradigms. While we empirically find little work in the SD-RL space, there is a larger trend to build software that integrates machine learning and complex simulation models (Alber et al., 2019; Peng et al., 2021). In science-focused endeavors, Hucka et al. (2018); Lang et al. (2020) aim to standardize the description of systems biology models in order to build simulation tools. The application of RL in domains such as fluid dynamics and supply chains is also noteworthy (Pirmorad et al., 2021; Giannoccaro & Pontrandolfo, 2002). Within SD, Thomas (2020); Rahmandad & Fallah-Fini (2008) look at incorporating RL algorithms to learn optimal policies, however most other uses of ML focus on parameter estimation and model selection (Gadewadikar & Marshall, 2023; Chen et al., 2011). Our work is positioned uniquely, sharing the same motivations as above, but focusing instead on generalized approaches to incorporate SD into RL.

Our research direction aims to address the well-known challenge of the lack of connection between benchmark models and the real world. Our work could easily be extended to turn the hundreds of standardized systems models, from SD and beyond, into RL environments.

## 6 Conclusion

In this paper, we introduced a proof-of-concept library, SDGym, that serves as a bridge between SD and RL. We began by navigating the paradigmatic differences between SD and RL and exploring the unique opportunities that their integration offers. We proceeded to frame SD intervention discovery as an optimization

problem for the RL setting. Finally we discussed design decisions that would enable a generalized approach to translating SD models into RL environments.

SDGym is a low-code library enabling seamless integration of SD models into the RL setting. We demonstrated its capabilities by initializing an RL environment using an example SD model, and training an agent to learn an optimal policy. Our demonstration underscored the potential of the SD-RL integration. Training a simple agent that can achieve specific goals validated that a well-specified RL environment can be built entirely from an SD model. We posit that our approach for environment construction will lead to more robust interventions that outpace the conventional methods in both efficacy and time-efficiency.

For RL practitioners, the integration offers a path to harness the rich domain-specific knowledge embedded in SD models, allowing for a better informed environment design through participatory methods. Simultaneously, SD practitioners stand to benefit immensely from the adaptive strategies and dynamic interventions made possible by RL, potentially redefining the scope of interventions and system modifications traditionally considered within the SD realm.

## 6.1 Future work

This paper sets the foundation for interesting work in a few areas. At the forefront is the potential role of SD in ML fairness, safety and robustness. Community-built SD models with rich context are critical to these responsible AI goals, providing an ability to simulate edge cases and navigate nuanced environmental conditions. Combination of SD and RL can assist in evaluating policies under various scenarios and optimizing potential outcomes. This potential needs to be assessed in the training and evaluation of ML models and RL agents in high-stakes domains like healthcare. Studies need to be conducted to understand the effectiveness of our approach compared to other causality-based approaches to robustness and assistive approaches to RL environment design.

The utility of SDGym for SD practice also needs to be investigated further, by comparing it with prevailing optimization methodologies within SD to evaluate its performance. Potential directions include calibration optimization and addressing high-dimensional, multi-objective, multi-agent societal challenges like the Sustainable Development Goals (SDGs). In the overarching SD and RL continuum, we foresee a symbiotic relationship where insights from RL agents can offer refinements to foundational SD models.

In closing, SDGym exemplifies the potential of integrating SD and RL in both directions. The scope and diversity of applications hold a lot of promise to accelerate innovation at the intersection of both fields.

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
