# OpenReview forum: "SDGym: Low-Code Reinforcement Learning Environments using System Dynamics Models"
_TMLR — Withdrawn by Authors_

### Review · Reviewer_edJU · 2023-11-07

**Summary Of Contributions:**

The paper proposes a library that converts sytem dynamics (SD) models into reinforcement learning (RL) environments. The contributed approach is validated by converting a specific SD model, dedicated to electric vehicles adoption, into a reinforcement learning environment. Finally, the paper shows that an RL agent can successfully learn in the new environment.

**Audience:**

No

**Broader Impact Concerns:**

I don't foresee broader impacts worthy of concern.

**Claims And Evidence:**

No

**Requested Changes:**

The changes I suggest are critical for acceptance.

- The paper should either remove the motivation from Responsible AI if there is no serious experimental assessment of the value of the approach in Responsible AI;
- The approach would be significant if it was shown how much easier is to use an SD model compared to building an RL environment from scratch, and that RL is, compared with other approaches, beneficial for typical SD problems;

Minor suggestions are the following.
- The paper should remove the irrelevant reinforcement learning background;
- The paper should explain more clearly SD models.

**Strengths And Weaknesses:**

Strengths:

- Joining the expertise of SD and RL research is possibly a relevant approach to solving important complex real-world problems. As the paper rightfully points out, existing SD models of complex real-world problems can be used to build RL environments from where RL agents can learn to act. The paper makes a good argument on this possible benefit. The proposed approach is clearly explained and in detail and the proposed validation for the approach is sound. Finally, the ideas and the execution are also novel as far as I know.

Weaknesses:

- I believe the paper could make a much clearer and stronger argument for the motivation of the work. It is mentioned that the work can be used for Responsible and Fair AI, but not much is done in showing the real benefits of the approach towards these ends. In fact, I miss specific definitions and example of how the proposed approach will help, and thorough experiments that compare the approach with others on these matters. I believe, that, with the current experiments and lack of detail, the paper would be better positioned by solely motivating the work by arguing that SD models are rich and we can can us the richness to have richer RL environments. TOn the other hand, there is not enough detail about SD. Specifically, the paragraph that is written has terms that are not clearly defined, and the picture in figure 1 is also not clear. Figure 2 is not readable, as the words are too small.
- On top of the misleading motivation, I believe the paper has irrelevant RL background. Specifically, sections 2.2.1, 2.2.2, and 2.2.3 are not necessary. On the other hand, the implementation, section 3, has too much information. I believe it is not necessary to describe the classes that were built on the code, as they do not had much on a higher-level, especially since the code should be made publicly available. Having said that, the contribution of the work is limited. While the idea is interesting, I believe the work would only be relevant for publication if there is a thorough experimental evaluation towards the ends defined. In other words, I believe that the approach is only relevant for publication if it addresses a specific problem and shows that (i) not using a pre-defined SD model and then the proposed approach would be wore than building the RL environment from scratch, and (ii) reinforcement learning performs better than other approaches on the created environment. Just showing that the code works in transferring an SD model into an RL environment, and that RL can learn on the environment, is not significant enough.

---

### Review · Reviewer_eDdW · 2023-11-08

**Summary Of Contributions:**

This paper presents a software package that implements System Dynamics specifications as a simulator with an RL interface.

**Audience:**

Yes

**Claims And Evidence:**

Yes

**Requested Changes:**

- It would be nice to have a figure showing a simple, most bare-bone SD model represented as (i) pictorially like Fig 1, (ii) in equations, (iii) in code according to the API of the proposed SDGym package. This is necessary to understand how hard or easy it is to use the proposed package.
- The authors are encouraged to include other sample / benchmark SD environments as part of this paper or the package documentation.

**Strengths And Weaknesses:**

Strengths:
- Flexible state, action, and reward definitions

Weaknesses:
- Only one example of EV adoption is shown in experiments.

---

### Review · Reviewer_Nobu · 2023-11-11

**Summary Of Contributions:**

This work integrates reinforcement learning (RL) framework into system dynamics (SD) framework through
the modifications of Open AI Gym environment.
It compares the difference of formulations of those two fields and shows how the RL can be used to optimize for
interventions etc. over SD.

**Audience:**

Yes

**Broader Impact Concerns:**

Not in the current work.

**Claims And Evidence:**

Yes

**Requested Changes:**

1. There are some typos.
For example, in page 3, agents' are strangely typed.  In page 8 top paragraph, "past" is strangely typed.
2. In page 3, 2.2.2 you mention "While potentially requiring more samples,..." can you elaborate on why this is the case?
3. While Table 1 is mentioned to be a representation of traditional approach, some of them are a bit questionable; could you please elaborate a bit on this?
4. Reward function definition 3.6 is a bit hard to parse.  The sentence may not be consistent with the equation (4), can you check them again?

**Strengths And Weaknesses:**

Strengths
1. The paper is well written and is concise, clearly conveying the core ideas
2. Conceptually, using SD as an inductive bias of the RL environment and using RL as a possible analysis tool for SD are both interesting and are made possible by integrating both of them.
3. It casts the work as one of the attempts of other work aimed at using some dynamics environment as inductive bias for RL (it is clear where this work stands)

Weaknesses
1. I would like to see a bit more convincing examples where this SD inductive bias helps RL for, say robustness, interpretability etc., while RL helps SD for anlaysis etc.  There are some interesting experimental examples already, but can you think of some examples (even if they are toy model and simple examples) where the combinations of RL and SD shine?

---

### Note · Authors · 2023-11-24

**Comment:**

We would like to thank the reviewers for taking time for reviewing the paper and providing suggestions.

**Withdrawal Confirmation:**

I have read and agree with the venue's withdrawal policy on behalf of myself and my co-authors.